# HCNET: A Point Cloud Object Detection Network Based on Height and Channel Attention

**Jing Zhang \***, **Jiajun Wang, Da Xu and Yunsong Li**

School of Telecommunication Engineering, Xidian University, Xi'an 610100, China;
jiajunwang129@gmail.com (J.W.); 21011210308@stu.xidian.edu.cn (D.X.); ysli@mail.xidian.edu.cn (Y.L.)
* Correspondence: jingzhang@xidian.edu.cn; Tel.: +86-137-5990-1785

**Abstract:** The use of LiDAR point clouds for accurate three-dimensional perception is crucial for realizing high-level autonomous driving systems. Upon considering the drawbacks of the current point cloud object-detection algorithms, this paper proposes HCNet, an algorithm that combines an attention mechanism with adaptive adjustment, starting from feature fusion and overcoming the sparse and uneven distribution of point clouds. Inspired by the basic idea of an attention mechanism, a feature-fusion structure HC module with height attention and channel attention, weighted in parallel, is proposed to perform feature-fusion on multiple pseudo images. The use of several weighting mechanisms enhances the ability of feature-information expression. Additionally, we designed an adaptively adjusted detection head that also overcomes the sparsity of the point cloud from the perspective of original information fusion. It reduces the interference caused by the uneven distribution of the point cloud from the perspective of adaptive adjustment. The results show that our HCNet has better accuracy than other one-stage-network or even two-stage-network RCNNs under some evaluation detection metrics. Additionally, it has a detection rate of 30FPS. Especially for hard samples, the algorithm in this paper has better detection performance than many existing algorithms.

**Keywords:** autonomous driving; point cloud; object detection; attention mechanism; adaptive adjusted

## 1. Introduction

In the 21st century, automatic driving has gradually broken through the limitation of hardware, which has sped up its research process. Its application and popularization will bring great changes to human society, such as the transformation of public transportation, infrastructure, and urban appearance. Currently, there are numerous high-tech companies, automobile manufacturers, and startup companies working on autonomous-driving technologies in order to build a smarter and safer system.

As one of the key technologies of automatic driving, object detection has made great progress in recent years [1]. Before the deployment of automatic driving by LiDAR, the camera was the main sensor. Since the image data can perceive color information, it can play a vital role in tasks such as traffic light recognition. However, the lack of accurate three-dimensional information cannot meet the safety requirements of autonomous driving scenarios.

In order to meet the needs of three-dimensional information for target detection, new technologies represented by LiDAR have made major breakthroughs in the real-time acquisition of multi-grade three-dimensional spatial targets in recent years. This system can partially block through the woods to directly obtain high-precision three-dimensional information on the real surface, which cannot be replaced by traditional photogrammetry. Therefore, the sources of information of the autonomous driving perception system [2] are mainly images and point cloud data.

The high-precision point cloud data generated by LiDAR makes up for this shortcoming, as it can obtain more detailed target-shape and three-dimensional position information and provide a more robust perception effect. On one hand, how to use point cloud information more effectively has become a recent research hotspot; on the other hand, as point clouds are usually disordered, sparse, and uneven, the task of point cloud target detection still poses a great challenge.

In order to interpret and understand a 3D point cloud, deep learning has become the mainstream approach in the field of object detection. Based on the two-dimensional deep-learning research, including images and videos, researchers are gradually expanding the powerful tool of deep neural networks to 3D point clouds.

Due to the irregular format, one of the biggest challenges in designing learning algorithms is to develop efficient data structures to represent 3D point clouds. To better apply mature deep-learning tools to the point cloud field, most methods first convert the disordered point cloud data into a regular voxel grid and then use 3D convolution [3] for processing.

In order to directly process the original point cloud, PointNet [4,5] takes the lead in using point level multi-layer perceptrons (MLPs) and maximum pooling to ensure the invariance of the arrangement. Following that, a series of 3D deep-learning methods are based on PointNet.

As the most representative point cloud target-detection network, VoxelNet [6] lays the foundation for the voxel-based point cloud target-detection algorithms. However, due to the high computational cost of 3D convolution, it is difficult to run in real-time. Thus, Yan et al. introduced sparse 3D convolution to reduce the computational cost of the network in SECOND [7].

Later, a new network, PointPillars [8], proposed to use the concept of pillars to regularize the point cloud data, to use PointNet to learn the features in the pillars, and then to project the point cloud into a 2D pseudo image. However, due to the limited resolution, the division method of pillars would cause a certain degree of information loss.

This paper proposes a new point cloud object detection network, HCNet, based on a 2D convolutional [9] network. To avoid the high computational cost of 3D convolution [10,11] and reduce the coding and quantization loss of the PointPillars method, we propose a coding method that can generate multi-layer pseudo images based on the attention mechanism. Specifically, for the divided voxel grid, we first gather them at the same height to generate multiple pseudo images containing point cloud information of different heights. Through the dual attention weighting processing of the height dimension and the channel dimension, we merge multiple pseudo images into a feature map.

Simultaneously, to overcome the sparseness and uneven distribution of the point cloud, we propose an adaptively adjusted detection head that, first, compensates for the sparseness of the point cloud by supplementing the original feature information. A distance-based adaptive weighting strategy is proposed to overcome the uneven distribution of the point cloud, so that the network can focus on key information.

The main contributions of this paper are as follows:

- The proposed HC module: This avoids the loss of information caused by quantifying point cloud data while enhancing the ability to express features;
- A self-adjusting detection head: This overcomes the impact of the sparseness and uneven distribution of the point cloud on the object detection task to a certain extent;
- The proposed algorithm has an inference speed of 30 fps and is comparable to the performance of the most advanced methods.

## 2. Related Work

Traditional point cloud perception methods usually use Euclidean clustering [12] or region generation [13] to group point clouds, and use ground-culling filtering [14] and map filtering algorithms [15] to increase the detection accuracy. In recent years, with the rapid development of deep learning, the research of point cloud target detection, based on

deep learning, has also burst into new vitality. The existing deep-learning target-detection methods are divided mainly into single-stage networks and two-stage networks.

### 2.1. Two-Stage Network

The two-stage [16] method first proposes several regions [17] that contain objects, and then determines the category label of each proposal by extracting the features in the regions [18]. Chen [19] et al. have generated a set of high-precision 3D candidate frames from the BEV map and projected them onto the feature maps of multiple views. They then combined the regional features of multiple views to predict the oriented 3D bounding box. Shi et al. put forth a PointRCNN [20] framework. Specifically, they segmented the three-dimensional point cloud directly to obtain the front scenic spot, and then fused the semantic features and local spatial features to generate high-quality 3D bounding boxes. They performed pioneering work on 3D object detection using graph convolutional networks based on PointRCNN. The F-PointNET [21] uses existing 2D object detectors to generate 2D candidate regions of the object and then extracts 3D frustum proposals for each 2D candidate region. The PointNet is applied to learn the point cloud features of each three-dimensional frustum to realize the estimation of the 3D bounding box.

### 2.2. Single-Stage Network

The single-stage method [22] does not require region-proposal generation. It uses a single-stage network directly to predict the target category probability and return to the 3D bounding box [23] of the object. Yang et al. [24] discretized the point cloud with equally spaced units and, similarly, encoded the reflectivity to obtain a regular representation. They then applied a full convolutional network to estimate the target's position and heading angle. This method is not only superior to the single-stage methods (including VeloFCN [25], 3D-FCN [26], and Vote3Deep [27]) in precision, but it is also greatly improved in speed. Zhou et al. proposed a voxel-based end-to-end trainable framework named VoxelNet. They segmented the point cloud into equally spaced voxels and encoded the features within each voxel into 4D tensors. They connected a region proposal network to produce test results. Although this framework achieved good performance, this method is very slow due to the sparsity of voxels and 3D convolution. Later, Yan et al. used sparse convolutional networks [28] to improve inference efficiency. Based on this network, Lang et al. proposed a 3D object detector called PointPillars, which uses PointNet to learn point cloud features in vertical columns and encode the learned features into a pseudo image. Later, the pipeline [29] of 2D target detection is applied to predict the 3D bounding box.

### 2.3. Attention Mechanism

Human visual attention enables us to pay more attention to specific areas in the picture, then gradually adjust the focus, move to other specific areas, and finally, infer the information of the whole picture.

In the various elements given in Figure 1, the red area indicates where attention is more focused. People will pay more attention to people's faces, the title of the text, and the first sentence of the article. This visualization shows how human beings efficiently allocate limited attention resources when viewing an image.

In order to allow the computer to choose the information that is more critical to the current task goal from a large amount of information, researchers have transferred this idea to deep learning.

The attention mechanism [30] in a neural network is a mechanism that allows the network to learn to focus on key information and ignore irrelevant information. It can make the neural network [31] focus on a subset of its inputs. The attention mechanism module can be divided into spatial domain attention [32], channel domain attention [33], and hybrid domain attention [34]. These classic divisions inspired us to apply the attention mechanism to the feature fusion of multi-layer pseudo images. For example, Point-SENet [35] utilized the channel attention of the SE module to predict the 3D detection box. Point2Sequence [36]

proposed an attention-based sequence-to-sequence structure to aggregate the sequential features of multi-scale areas.

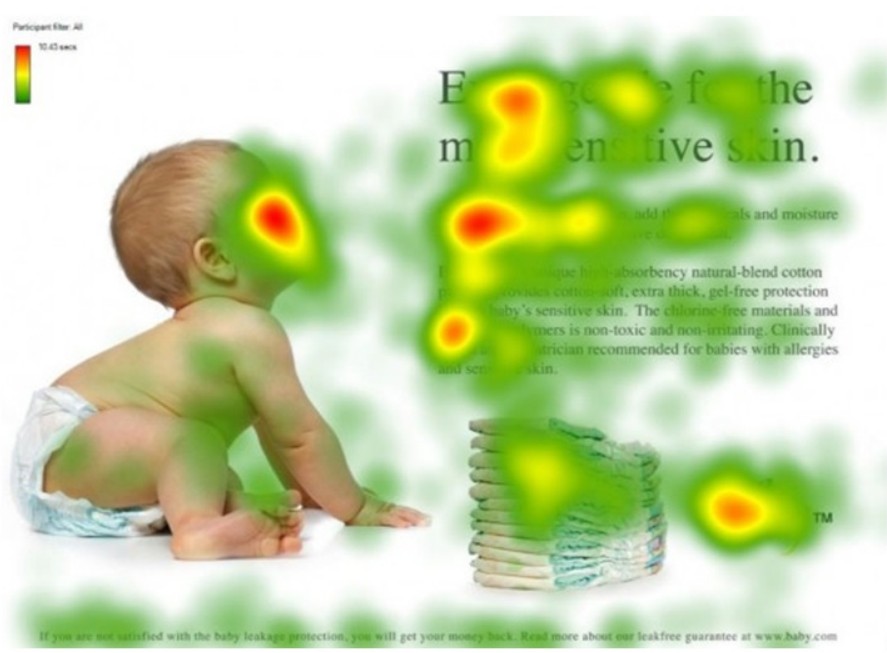

**Figure 1.** Human visual attention.

### 3. HCNet

In this section, we introduce HCNet, which outputs accurate classification and 3D bounding boxes based on the LiDAR point cloud.

The HCNet is a single-stage end-to-end trainable network. As shown in Figure 2, HCNet consists of three blocks:

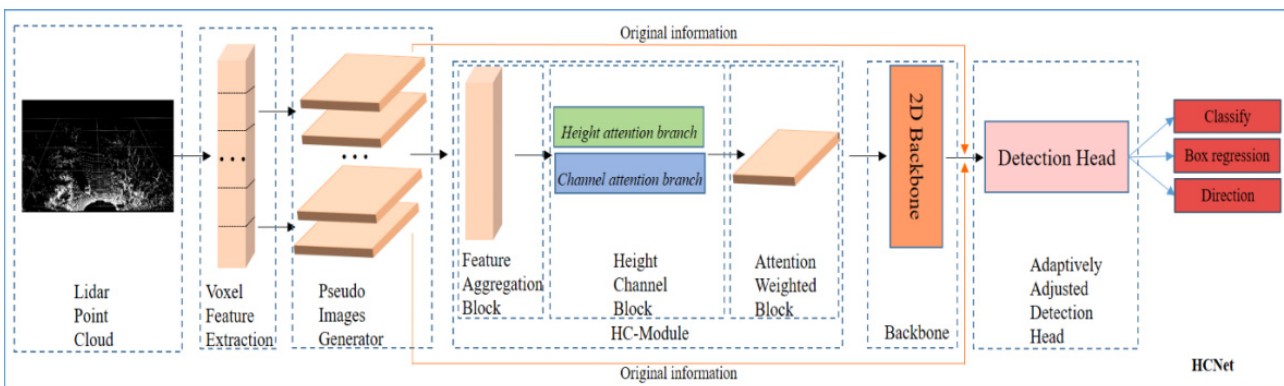

**Figure 2.** The structure of HCNet. The original point cloud is used as the input, and multiple pseudo images are fused by the HC module; in addition, the adaptive detector head is used to overcome the sparsity and uneven distribution of the point cloud.

(1) The HC module fully aggregating the original features of the point cloud through the dual attention of height dimension and channel dimension;

(2) a backbone network that uses a 2D CNN to extract features;

(3) an adaptive detection-head network that adjusts the feature map and then outputs the labels and bounding boxes of the objects.

### 3.1. HC Module

In consistence with the existing voxel-based methods, we first divide the point cloud space into voxel grids. For the three-dimensional point cloud space with the range of (D, H, W), we divide it multiple times at equal intervals from three dimensions. Firstly, it is divided into N and M times in the horizontal direction (distance), and then P times in the vertical direction (height). The obtained voxel size is D′ = D/N, H′ = H/M, and W′ = W/P. According to the division position, each voxel has a division coordinate (x, y, z). We used a strategy similar to PointNet to extract the features within the voxel and aggregated the features at the same height into pseudo images according to the coordinate z to generate pseudo images.

In Figure 3, taking P = 3 as an example, the amount of vertical cutting can be adjusted according to different needs.

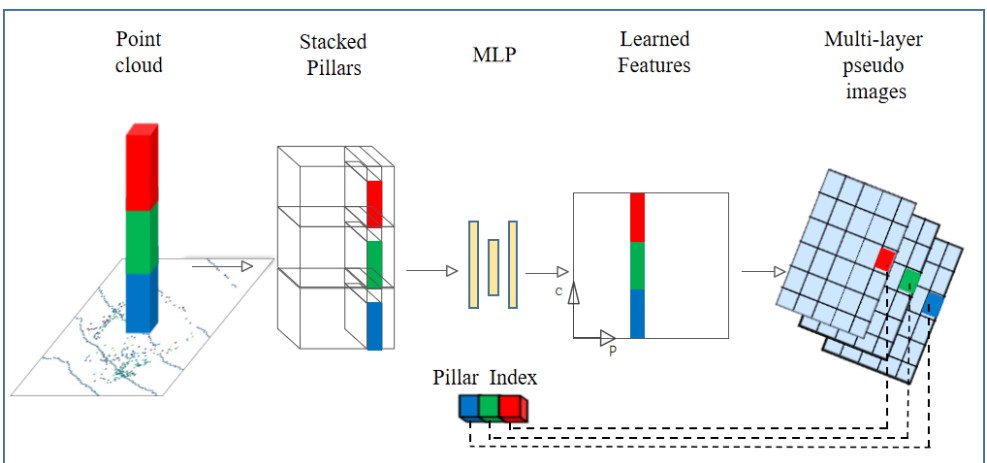

**Figure 3.** Pseudo images generator (P = 3).

The HC module aggregates multiple pseudo images by introducing an attention mechanism to focus on the point cloud features in different height ranges. Its structure is shown in Figure 4, which consists of three parts: Feature Aggregation Block, HC Block, and Attention-Weighted Block.

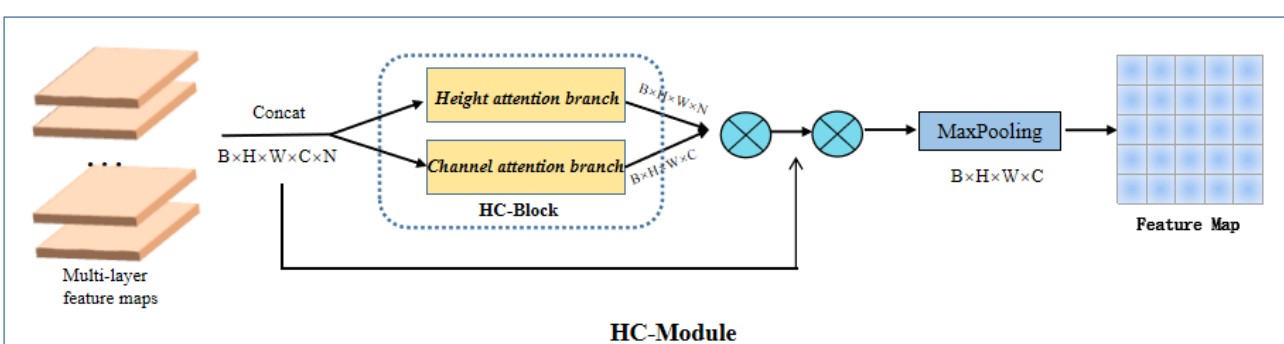

**Figure 4.** The structure of the HC module. It includes a height attention branch and a channel attention branch, which adjust features from two dimensions, respectively.

### 3.2. Feature Aggregation Block

The feature aggregation part is used mainly to aggregate the features of the pseudo images.

Here, the un-squeeze operation is used first to expand the size of each pseudo image to (B, C, H, W, 1), where B represents the batch size, H and W represent the height and width of the feature map, C represents the number of channels, and 1 represents the height

dimension from the extension. Then, we use the concat operation to splice the N feature maps to obtain a tensor P of size (B, C, H, W, N).

### 3.3. HC Block

The HC block realizes the fusion of multi-layer feature maps mainly through two parallel attention mechanisms. It contains two branches: a height attention branch (Figure 5) and a channel attention branch (Figure 6).

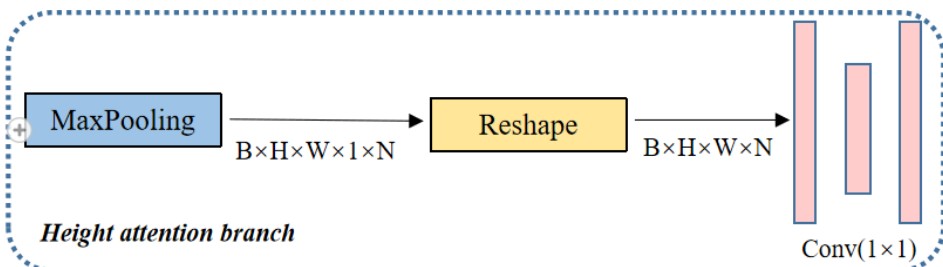

**Figure 5.** Diagram of the height attention branch.

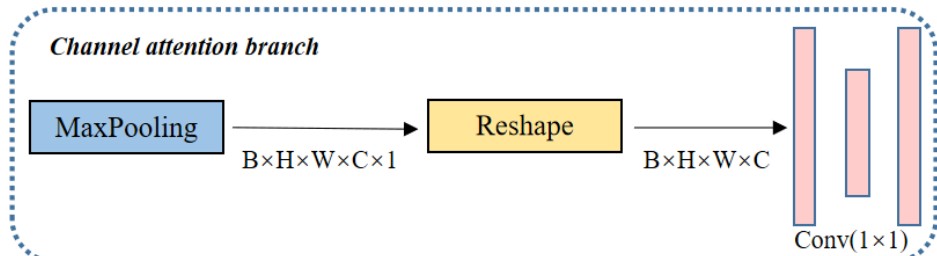

**Figure 6.** Diagram of the channel attention branch.

### 3.3.1. Height Attention Branch

Given a tensor *P*, we perform max-pooling to aggregate the features across channel-wise dimensions. The output size is (B, 1, H, W, N) for the channel-wise feature. Then, the reshape operation is used to generate an attention map $F_h$ of size (B, H, W, N). To explore the feature correlation between different heights, the height attention branch uses a convolution module to generate the final height global response. Assuming that $F_{in}$ is the feature map output by the max-pooling operation, the height attention branch can be expressed by (1). The size of the output $F_{att}^h$ will be (B, H, W, N).

$$F_{in} = \max(P)$$
$$F_h = \text{Re}shape(F_{in})$$
$$F_{att}^h = \text{Re}lu(Conv_{att}^h(F_h)) \tag{1}$$

Here, the operation of max($\cdot$) means that the max-pooling operation is performed to aggregate the height features across their channel-wise dimensions, the operation Re*shape*($\cdot$) is for reshaping the feature map into (B, H, W, N), Re*lu*($\cdot$) represents the ReLU activation function, and *Conv*($\cdot$) represents the convolution module, which contains two 1 × 1 convolutional layers.

### 3.3.2. Channel Attention Branch

The role of the channel attention branch is to enable the network to focus on more important channel features and reduce focus on unimportant channels. Primarily, given the input tensor *P*, it first needs to pass the max-pooling operation to complete the feature aggregation by height and output the feature map of size (B, C, H, W). Then, a global

channel response is also generated through the convolution module. The channel attention can be formulated as represented in (2). The size of the $F_{att}^c$ would be (B, C, H, W).

$$F_{in} = \max(P)$$
$$F_c = \text{Re}shape(F_{in})$$
$$F_{att}^c = \text{Re}lu(Conv_{att}^c(F_c)) \tag{2}$$

Here, the operation of $\max(\cdot)$ means that the max-pooling operation is performed to aggregate the height features across their height-wise dimensions, the operation $\text{Re}shape(\cdot)$ is for reshaping the feature map into (B, H, W, C), $\text{Re}lu(\cdot)$ represents the ReLU activation function, and $Conv(\cdot)$ represents the convolution module, which contains two $1 \times 1$ convolutional layers.

### 3.4. Attention-Weighted Block

The attention-weighted block mainly uses $F_{att}^h$ and $F_{att}^c$ to complete the attention adjustment of the feature tensor. It realizes the dual adjustment of the height dimension and the channel dimension, and then obtains richer feature information. A max-pooling operation is used to aggregate the feature from the backbone. The attention-weighted block can be formulated as given in (3). The size of the $F_{out}$ will be (B, C, H, W).

$$F_{att}^{hc} = sigmoid(F_{att}^h \otimes F_{att}^c)$$
$$F = F_{att}^{hc} \cdot P$$
$$F_{out} = \max(F) \tag{3}$$

Here, $\otimes$ represents the matrix multiplication, $sigmoid(\cdot)$ represents the sigmoid function employed to normalize the values of the attention matrix to a range of (0,1), the second equation represents the element-wise multiplication (that is, the completion of the tensor P in the height and channel dimensions' parallel adjustment), and $\max(\cdot)$ means the max-pooling operation preformed to aggregate the features across their height-wise dimensions.

### 3.5. Backbone

We use a similar backbone as the YOLO [37], and the structure is shown in Figure 7.

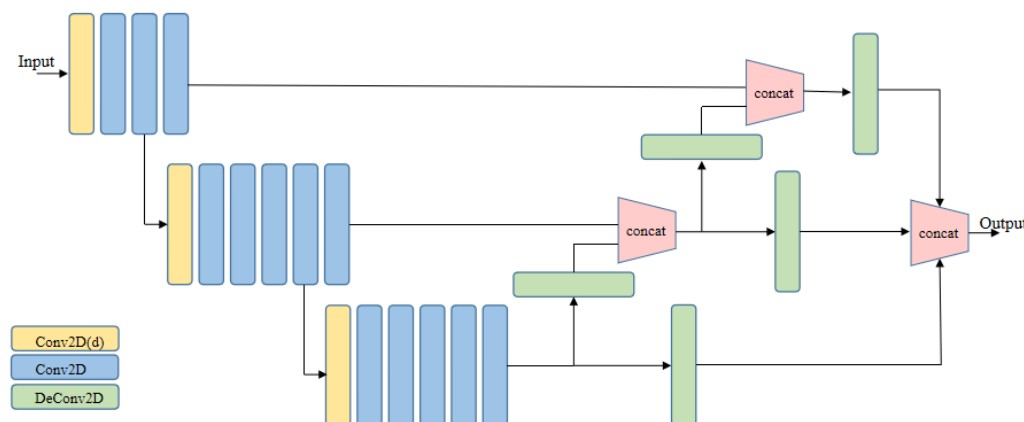

**Figure 7.** Diagram of the backbone network.

The backbone has three parts:

(1) the top-down part, which generates features at smaller and smaller resolutions;
(2) the down-top part, which up-samples the feature map from the bottom up and stitches it with the features of the upper layer;
(3) the third part, which up-samples the features of each layer of the network to the same size through up-sampling, and then splices to generate the final output features.

The top-down part consists of a series of convolutional blocks; each block contains a convolutional layer with a convolution kernel size of 3 × 3, each followed by BatchNorm and a ReLU. The down-top part comprises a series of deblocks. Each deblock contains a 2D transposed convolution with a 3 × 3 convolution kernel used to adjust the size of the current feature map to the same size as the previous feature. The final feature maps from each down-top deblock are combined through up-sampling and concatenation to generate the input of the detection head.

### 3.6. Detection Head

In this paper, we propose an adaptively adjusted detection head that enables the network to compensate for the detailed information and adaptively adjust the sparse feature map, thereby improving the object detection performance of the network structure.

This section proposes an adaptively adjusted detection head that includes three parts: a detection head backbone network, an original information-fusion module, and an adaptive-adjustment module. The network structure in this section is presented in Figure 8. The original information-fusion module mainly weakens the sparsity of the point cloud from the perspective of compensating for information, whereas the adaptive-adjustment module implements adaptive adjustment on the feature map to avoid the uneven distribution.

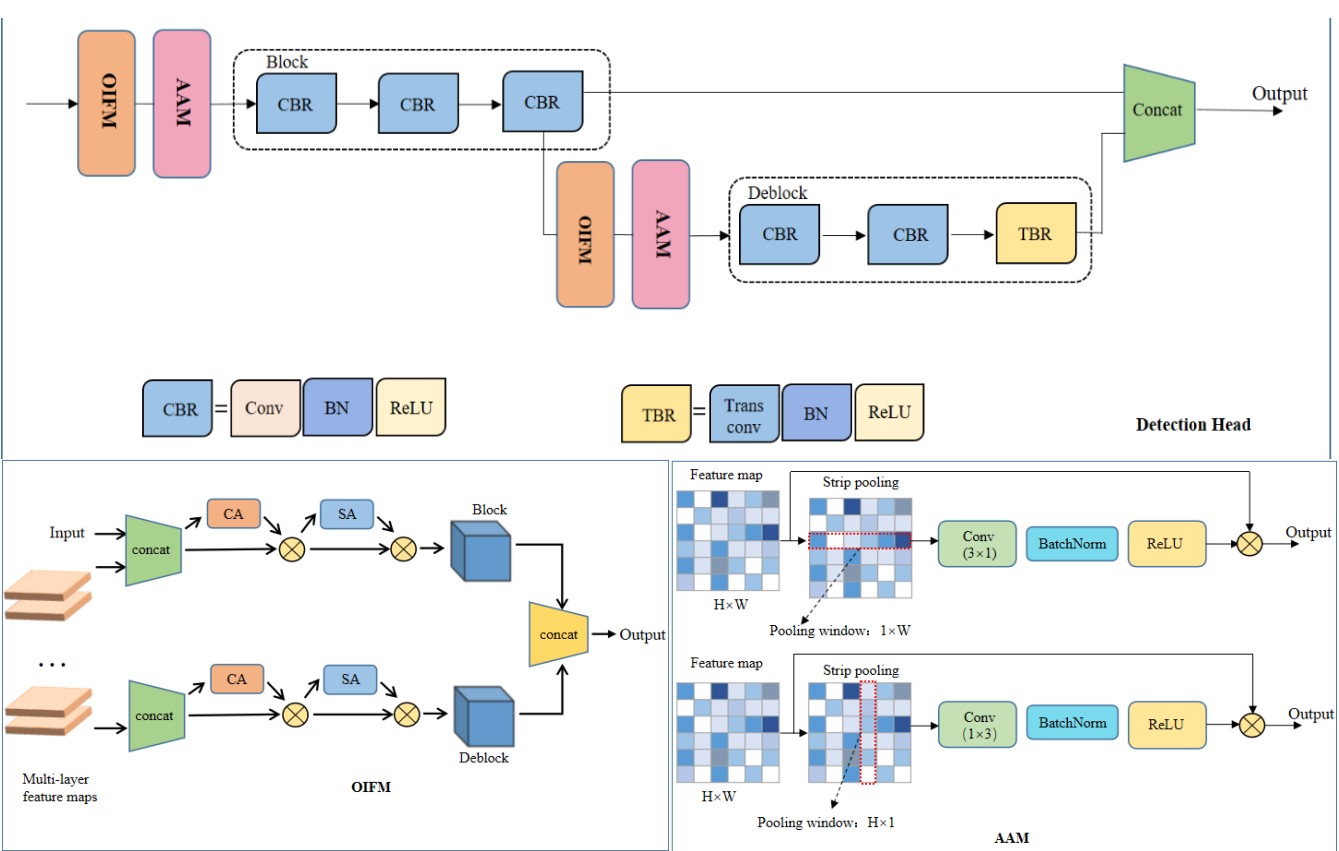

**Figure 8.** The structure of the adaptively adjusted detection head. The structure of the OIFM (original information fusion module) is shown in the lower-left part of the figure, and the structure of the AAM (adaptive adjustment module) is shown in the lower-right part of the figure.

#### 3.6.1. Detection Head Backbone Network

The main frame of the detection head, designed in this section, uses a top-down multi-scale network, which can enhance the ability of the network model through the fusion of information at different scales.

As shown in Figure 7, the backbone network consists of a block module and a deblock module, and the results of the two modules are aggregated through the concat operation.

Thus, the network has features that contain different receptive fields. By merging these multi-scale features, a higher quality target-prediction regression effect is achieved.

### 3.6.2. Original Information-Fusion Module

This mainly incorporates the features of the original point cloud pseudo images into the input features of the block and deblock modules of the backbone network of the detection head. Then, the merged features are adjusted from the channel dimension and space dimension.

First, through the convolution operation, the sizes of the two layers of the pseudo feature maps are adjusted to the same size as the input features of the block module and the deblock module, and the information is then aggregated through the concat operation. A similar strategy is also used by CBAM [38] to perform feature weighting.

### 3.6.3. Adaptive-Adjustment Module

The adaptive-adjustment module adjusts the width and height of the feature map so that the information in the key area of the feature map can be emphasized. Thus, the network can pay attention to the sparse object point cloud information from a distance. First, we aggregate the features through strip pooling. For an input feature map $F_{in} \in R^{H \times W}$, with height H and width W, the pooling-window size of the strip pooling is usually $H \times 1$ or $1 \times W$. While performing feature aggregation in the width direction of the feature map, its pooling window should be set to $1 \times W$, and the specific pooling process can be expressed as (4).

$$y_i^h = \frac{1}{W} \sum_{0 \le j \le W} x_{i,j} \tag{4}$$

Here, $y_i^h$ is the feature pooled by the horizontal pooling window representing the aggregate feature of the current row.

Similarly, when performing strip pooling in the height direction of the feature map, the pooling window should be set to $H \times 1$, and the specific calculation process is as shown in (5).

$$y_j^w = \frac{1}{H} \sum_{0 \le i \le H} x_{i,j} \tag{5}$$

Here, $y_j^w$ is the feature pooled by the vertical pooling window, which represents the aggregate feature of the current column.

Through the horizontal and vertical strip-pooling operations, the features in the strip area can be coded collectively, and the key, detailed features can be captured at the same time. We thus generate row-compression information of size $H \times 1$, and column-compression information of size $1 \times W$.

Then, the convolution component is used to extract the adaptive information, in both horizontal and vertical directions, from the two sets of obtained compressed information. The basic structure of the convolution component comprises a 2D convolution layer, a BatchNorm layer, and a ReLU function layer. Consequently, the weight adjustment of the feature map is realized by multiplying element by element. $F_{in}$ and $F_{out}$, respectively, represent the adaptively adjusted input and output feature maps. $F_{in}^{strip}$ represents the feature map after the strip-pooling process. $F_a^{strip}$ represents the feature map after convolution and up-sampling.

The overall calculation process can be expressed as (6).

$$\begin{aligned} F_{in}^{strip} &= strip(F_{in}) \\ F_a^{strip} &= up(Conv(F_{in}^{strip})) \\ F_{out} &= F_a^{strip} \cdot F_{in} \end{aligned} \tag{6}$$

In the above formula, $strip(\cdot)$ represents the strip pooling process. The function $Conv(\cdot)$ uses convolution to extract the hidden information from the pooled features, while the function $up(\cdot)$ represents an up-sampling operation used to generate global adaptive information of the same size.

### 3.7. Loss Function

We use the same loss function as SECOND. This further consists of three different kinds of loss functions: regression loss, classification loss, and directional loss.

Regression loss function—In 3D target detection, a 3D bounding box is usually defined by the parameters $(x, y, z, h, w, l, \theta)$. The residual positioning coding of the ground truth and anchors are defined by (7).

$$
\begin{aligned}
&\Delta x = \frac{x^{gt} - x^a}{d^a}, \Delta y = \frac{y^{gt} - y^a}{d^a}, \Delta z = \frac{z^{gt} - z^a}{d^a} \\
&\Delta w = \log \frac{w^{gt}}{w^a}, \Delta l = \log \frac{l^{gt}}{l^a}, \Delta h = \log \frac{h^{gt}}{h^a} \\
&\Delta \theta = \sin(\theta^{gt} - \theta^a), d^a = \sqrt{(w^a)^2 + (l^a)^2}
\end{aligned}
\tag{7}
$$

The regression loss function is expressed as (8).

$$
L_{loc} = \sum_{b \in (x,y,z,w,l,h,\theta)} SmoothL1(\Delta b)
\tag{8}
$$

Classification loss function—Similar to the existing methods, we also use the focal loss to handle the class imbalance problem. The classification loss is formulated as (9):

$$
L_{cls} = -\alpha_a (1 - p^a)^\gamma \log p^a
\tag{9}
$$

where $P^\alpha$ is the class probability of an anchor. We use the original paper's settings $\alpha = 0.25$ and $\gamma = 2$.

Directional loss function—To enable the network to learn the direction information of the bounding box, SoftMax is used as the direction loss function $L_{dir}$.

By combining all the loss functions, the total loss is defined as (10).

$$
L = \frac{1}{N_{pos}} (\beta_{loc} L_{loc} + \beta_{cls} L_{cls} + \beta_{dir} L_{dir})
\tag{10}
$$

Here, $N_{pos}$ represents the number of positive matching boxes, and the loss weights $\beta_{loc}$, $\beta_{cls}$, and $\beta_{dir}$ are empirically set to 2.0, 1.0, and 0.2, respectively.

## 4. Implementation Details

### 4.1. Dataset

All experiments in this study are tested with the KITTI [39] dataset as a benchmark. It contains real image data collected from city scenes, rural scenes, road scenes, and so on. There are up to 15 vehicles and 30 pedestrians in each image, as well as various degrees of occlusion and truncation (Figure 9).

KITTI has three categories: cars, pedestrians, and bicycles. Our dataset also contains object labels in the form of 3D tracklets, and we provide online benchmarks for stereo, optical flow, object detection, and other tasks.

The KITTI dataset comprises 7481 training samples and 7518 test samples. Each training sample contains cloud data and the corresponding RGB image data. Additionally, for each target category, there are three difficulty levels: easy, medium, and difficult. In this work, we focus only on the most challenging car categories.

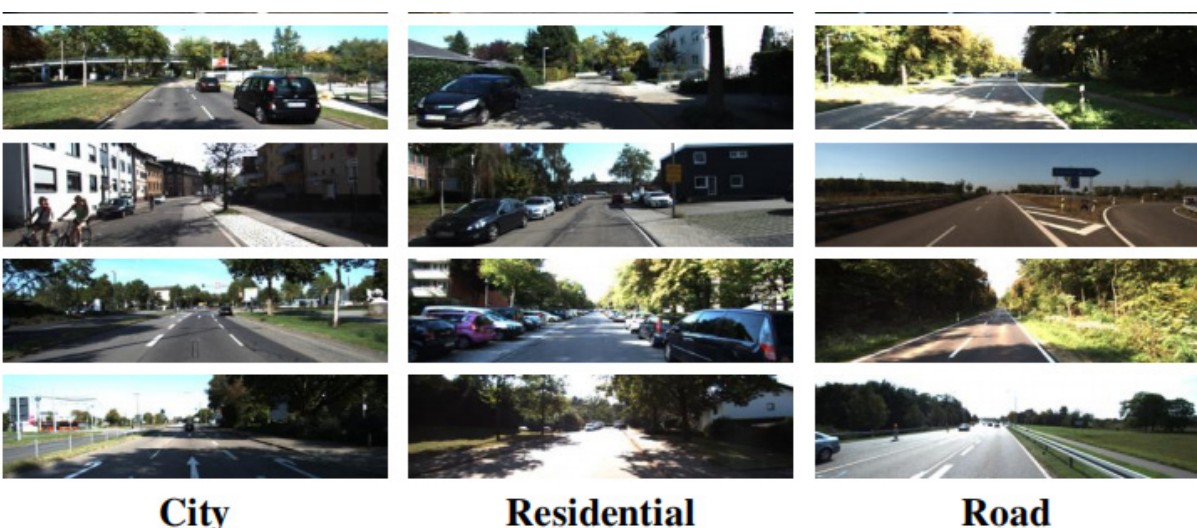

**Figure 9.** Examples from the KITTI dataset [39].

*4.2. Setting*

We set the range of (D, H, W) to ((0, 70.4), (−40, 40), (−3, 1)), and the number of voxel divisions, in the three directions, to N = 496, M = 432, and P = 4 to generate voxels of size 90.16, 0.16, 10. The car anchor has a (width, length, height) of (1.6, 3.9, 1.5) m. additionally, matching uses positive and negative thresholds of 0.6 and 0.45.

*4.3. Data Augmentation*

Data augmentation is essential to obtain good performance on the KITTI dataset. Inspired by PointPillars, our expansion strategy is as follows.

First, a database is generated from the training dataset containing the complete ground truth and their corresponding point cloud data.

Then, 15 ground truth samples are selected randomly from this database during the training process and introduced into the current training point cloud.

Moreover, to consider the influence of noise, we need to transform each ground truth and point cloud datum independently and randomly. Each ground truth is rotated randomly. The degree of rotation obeys a uniform distribution ($-\pi/2$, $\pi/2$).

Finally, the entire point cloud performs global scaling and a rotation operation. The degree of scaling obeys the uniform distribution (0.95, 1.05), whereas the degree of rotation obeys the uniform distribution ($-\pi/4$, $\pi/4$).

## 5. Experiments

*5.1. Evaluation Criteria*

This section describes the Intersection-over-Union (IoU), precision rate (Precision), recall rate (Recall), and average accuracy (AP). The IoU measures the degree of coincidence between the network prediction target box and the original real target box, which can be expressed in (11), which is the ratio between the intersection between the prediction box and the prediction box and the real box.

$$IoU = \frac{\mathrm{Detectionresult} \cap \mathrm{GroundTruth}}{\mathrm{Detectionresult} \cup \mathrm{GroundTruth}} \tag{11}$$

The role of the IoU is to help resolve the correctness of the detection results, and the general algorithm sets a threshold of T that is considered correct when the IoU > T is otherwise judged incorrect. On this basis, the prediction results can be divided into the following four categories:

(1)   TP (True positives): Positive samples were identified as positive samples;
(2)   TN (True negatives): Negative samples were identified as negative samples;

(3) FP (False positives): Negative samples were misidentified as positive samples;

(4) FN (False negatives): Positive samples were misidentified as negative samples.

On the basis of the IoU, the definitions of Precision and Recall can be given. The calculation process of Precision is shown in (12), which is actually among all the predicted targets, predicting the correct ratio.

$$Precision = \frac{TP}{TP + FP} \tag{12}$$

The computational procedure of Recall, as shown in (13), is the proportion of samples correctly identified as positive samples out of all positive samples in the test set.

$$Recall = \frac{TP}{TP + FN} \tag{13}$$

AP, which can be calculated using Precision and Recall, is actually the area under the Precision–Recall curve. Generally speaking, the better the target detection model is, the higher its AP value will be. To contrast this with existing algorithms, AP is also used as the main detection index.

### 5.2. Evaluation Details

We evaluated our HCNet on the KITTI validation set and the test set, to verify the superiority of our proposed model that only uses point clouds for 3D target detection, in Table 1.

**Table 1.** The comparison of performances in 3D object detection AP (%) on the KITTI validation set for the class of car.

| Method | Times(s) | 3D | | |
|---|---|---|---|---|
| | | Easy | Moderate | Hard |
| VoxelNet [6] | 0.23 | 81.97 | 65.46 | 62.85 |
| SECOND [7] | 0.05 | 85.50 | 75.04 | 68.78 |
| PointPillars [8] | 0.016 | 87.50 | 77.01 | 74.77 |
| PointRCNN [20] | 0.10 | 88.26 | 77.73 | 76.67 |
| HCNET (ours) | 0.032 | **88.45** | **78.01** | **76.72** |

We compare our HCNet with other existing algorithms performed on the KITTI validation set (Figure 10).

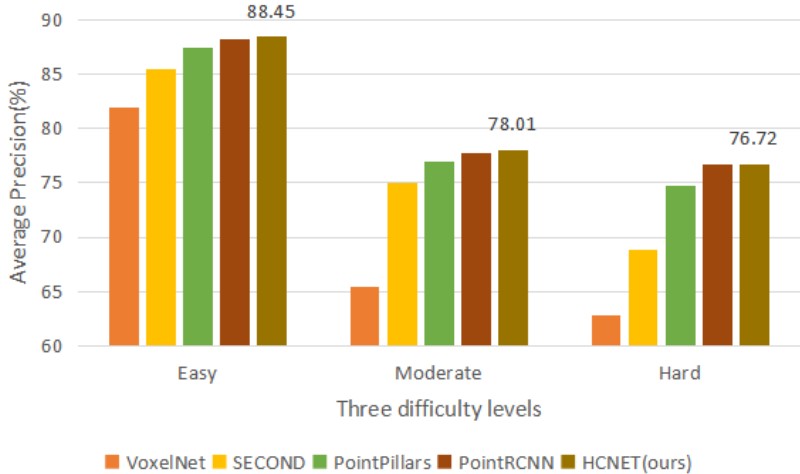

**Figure 10.** Performance of different algorithms (3D).

The algorithms to be compared include not only classic single-stage methods such as VoxelNet, SECOND, and PointPillars, but also two-stage methods with superior performance, such as the PointRCNN. Moreover, to compare our HCNet with other methods, we also submitted the results to the official KITTI server. The result of BEV and 3D are given in Table 2.

**Table 2.** The comparison of performances in bird's eye view (BEV) detection AP (%), 3D object detection AP (%), and object orientation estimation AOS (%) on the KITTI test set for the class of car.

| Method | Times(s) | BEV | | | 3D | | | Orientation | | |
|---|---|---|---|---|---|---|---|---|---|---|
| | | Easy | Moderate | Hard | Easy | Moderate | Hard | Easy | Moderate | Hard |
| VoxelNet [6] | 0.23 | 89.35 | 79.26 | 77.39 | 77.47 | 65.11 | 57.73 | N/A | N/A | N/A |
| Second [7] | 0.05 | 88.07 | 79.37 | 77.95 | 83.13 | 73.66 | 66.20 | 87.84 | 81.31 | 71.85 |
| PointPillars [8] | 0.016 | 88.35 | 86.10 | 79.83 | 79.05 | 74.99 | 68.30 | 90.19 | 88.76 | 86.38 |
| PointRCNN [20] | 0.10 | 89.28 | 86.04 | 79.02 | 84.32 | 75.42 | 67.86 | 90.76 | 89.55 | 80.76 |
| SARPNET [40] | 0.05 | 88.93 | **87.26** | 78.68 | **84.92** | **75.64** | 67.70 | 90.16 | 88.86 | 80.05 |
| HCNET (ours) | **0.032** | **90.27** | 86.91 | **81.90** | 81.31 | 73.56 | **68.42** | **95.05** | **91.44** | **88.02** |

It is worth noting that our method achieved very good results on the Hard difficulty level, and achieved good performance on both BEV and 3D. For BEV, we achieved comparable performance to SAPRNET on the Moderate (Mod) difficulty level, and achieved the best performance on the Easy and Hard difficulty levels. For 3D, we demonstrate good performance on the Hard level, but the results are poor for the Easy and Mod levels. The possible reasons are as follows:

(1) Our HC module focuses on more challenging tasks;
(2) Our adaptive detection head overemphasizes the features of difficult objects in the distance.

In addition, our method has also achieved excellent results on AOS, which shows that our algorithm also has strong performance in the prediction of orientation (Figure 11), which is very critical for autonomous driving tasks.

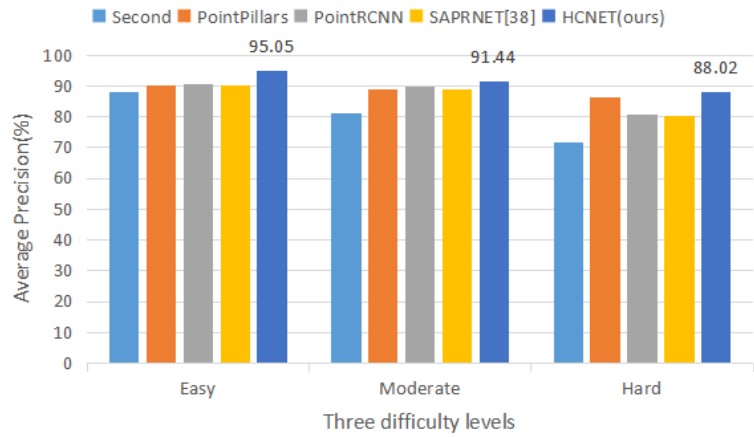

**Figure 11.** Performance of different algorithms (AOS).

We conducted multiple ablation experiments on the KITTI validation set to analyze the performance of different components of the HCNet. The training set used is the KITTI training set, the test set is the KITTI validation set, and the experimental platform is a computer with a GTX 2080Ti.

### 5.2.1. Different Attention Mechanisms in the HC Module

We compared the performance of multiple attention branches in the HC block on 3D target detection tasks. Channel attention (CA), height attention (HA), and height–channel attention (HCA) are shown in the Table 3.

**Table 3.** The comparison of the results of different attention mechanisms in HC module on the KITTI validation set for the class of car.

| Method | 3D | | |
|---|---|---|---|
| | **Easy** | **Moderate** | **Hard** |
| CA | 87.22 | 77.33 | 75.78 |
| HA | 87.19 | 77.26 | 75.82 |
| HCA | **88.45** | **78.01** | **76.72** |

It can be seen from Figure 12 that when fusing multiple feature maps, channel attention achieved better performance than high attention. Additionally, the HCA, which combines the two attention mechanisms in parallel, achieved the best detection results. This observation shows that the HC module we proposed can focus on the key features through weighted adjustments to the two dimensions of channel and height. By improving the information-expression ability of the feature map, the object-detection performance of the network is also improved.

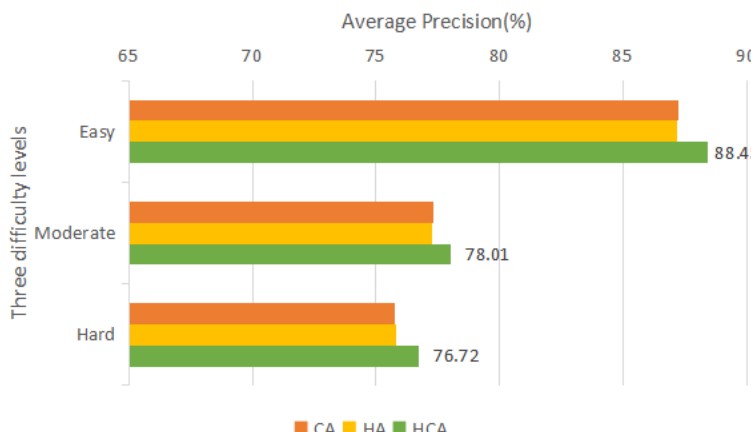

**Figure 12.** Performance of different attention mechanisms.

### 5.2.2. Different Parts of the Detection Head

We have the original information-fusion module (OIFM) and the adaptive-adjustment module (AAM) in the adaptive detection head. Ablation experiments were carried out to test their effect on the detection results.

From the results in the Table 4, it can be seen that both the OIFM and AAM help improve the detection effect, and the effect of OIFM is even more prominent. Additionally, the best detection performance can be achieved by connecting two modules serially. These facts show that our adaptive detection head overcomes the sparseness of the point cloud through the fusion of original information and makes up for the uneven distribution of the point cloud through the adaptive adjustment of the feature map in two directions.

**Table 4.** The comparison of the results of different parts in the detection head on the KITTI validation set for the class of car.

| Method | 3D | | |
| --- | --- | --- | --- |
| | **Easy** | **Moderate** | **Hard** |
| AAM | 87.23 | 77.33 | 75.70 |
| OIFM | 88.32 | **78.08** | 76.52 |
| (AAM + OIFM) | **88.45** | 78.01 | **76.72** |

*5.3. Visualization*

In order to show the detection results more visually, this section uses the MayaVi library to visualize the detection results of the network. MayaVi library is a general purpose, cross-platform tool for 3D scientific data visualization.

We selected two examples (Figures 13 and 14). They were obtained from the easy and hard datasets, respectively. The image on the left are the original point cloud, and the image on the right are the detection results. The white dots are formed by the LiDAR emission on the surface of the object, the green box is the target, and the number beside it is the confidence (Figure 15). Confidence means the probability that the current predicted box is the correct box.

As seen in the RGB diagram of Figure 13, this is a basic pavement scene. A good spacing remained between vehicles, with no occlusion and truncation to facilitate the measurement of the underlying performance of the algorithm.

The RGB of Figure 14 shows that various vehicles with different directions are parked on both sides of the road. High occlusion was highly truncated between each vehicle. This is very challenging for the target-detection task.

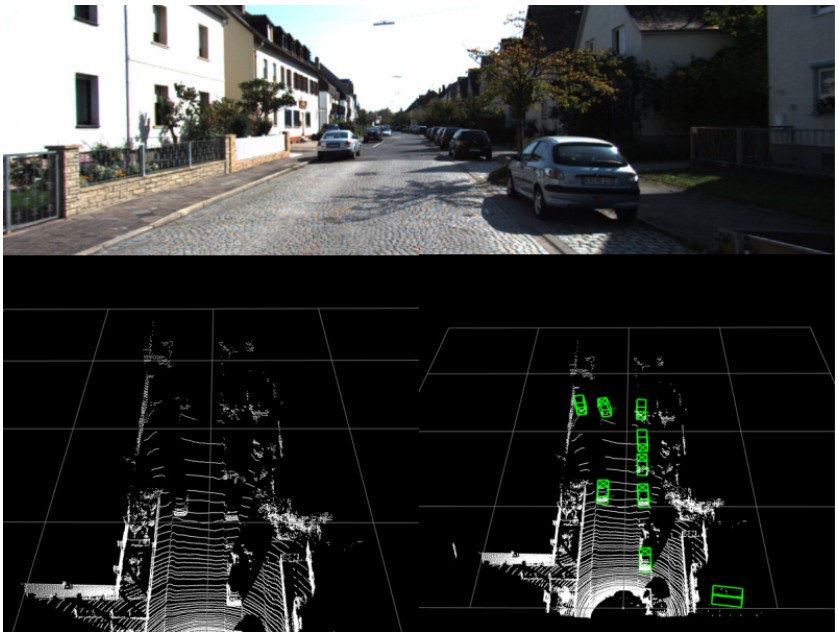

**Figure 13.** Example 1 (low occlusion and truncation).

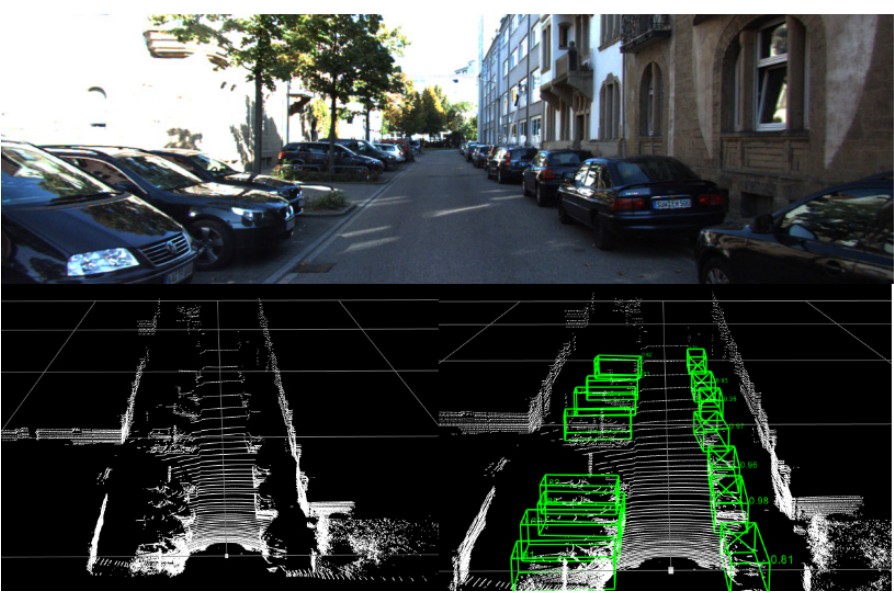

**Figure 14.** Example 2 (high occlusion and truncation).

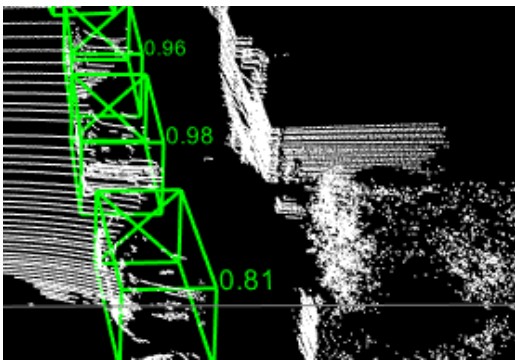

**Figure 15.** Confidence.

## 6. Conclusions

We propose a single-stage point cloud target-detection algorithm HCNet, and design the HC module through the attention mechanism to complete the feature fusion of multi-layer pseudo images and retain the original point cloud information to a greater extent. Using the parallel attention-mechanism module, the feature adjustment of the height dimension and the channel dimension is completed simultaneously, and the information-expression ability of the feature map is enhanced.

Moreover, we also designed an adaptively adjusted detection-head network for the sparsity of the point cloud and the uneven distributions of near-dense and far-field. On the one hand, the original information-fusion module is used to complete the information supplement of the feature map, thereby overcoming the sparsity of the point cloud; whereas, on the other hand, through the adaptive-adjustment module, the information in the key areas of the feature map are emphasized.

Thus, the network can pay attention to the target point cloud information that is scarce at a distance. The application of HCNet on KITTI shows that, compared to many existing algorithms, the HCNet proposed in this paper has achieved better detection results.

**Author Contributions:** Conceptualization, J.Z. and J.W.; methodology, J.Z. and J.W.; software, J.Z., J.W., D.X. and Y.L.; validation, J.W. and D.X.; formal analysis, J.Z.; investigation, J.W. and D.X.; resources, J.Z.; data curation, J.W. and D.X.; writing—original draft preparation, J.W.; writing—review and editing, J.Z., J.W. and D.X.; visualization, J.W. and D.X.; supervision, Y.L.; project administration, J.Z. and Y.L. All authors have read and agreed to the published version of the manuscript.

**Funding:** This research was supported by the Natural Science Foundation of China under Grant 61801359.

**Conflicts of Interest:** The authors declare no conflict of interest.

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
