# Peer review of "HCNET: A Point Cloud Object Detection Network Based on Height and Channel Attention"

_remotesensing, doi:10.3390/rs13245071_

Round 1

Reviewer 1 Report

The authors propose a method for object detection using point clouds. The method is based on height and channel attention.

The proposal is interesting. However, there is a number of issues that must be clarified to completely prove the validity of the approach and the interest to readers:

  • Every time that a new symbol appears in the paper, its meaning should be clearly explained. Also, the authors should avoid using the same symbol to represent different concepts. This is specially important in page 10, but the authors should carefully revise the whole paper.
  • In line 234, I guess the authors refer to C instead of N.
  • In sections 4 and 5, when the authors refer to methods by other authors to compare the performance of their proposal, they should include the reference where these methods are presented (e.g. in table 1, table 2).
  • In table 2 please highlight with bold font the best result for every column. Also, in this table, I guess the authors mean SARPNET instead of SAPRNET.
  • I consider it specially relevant to include some other classes of the Kitti dataset in the experimental section to prove the validity of the approach (cyclist and pedestrian). 
  • Also, the authors should include the results available on the Kitti project website of those recent methods that operate with point clouds (at least, those methods which have been published in recent years). This way, the reader can have a complete perspective of the performance of the method.

There are also some minor issues:

  • How have the authors ordered the references (e.g., reference [7] is cited in the text before reference [4])?.
  • In lines 70 and 73, Second and Pointpillars are the names of two approaches proposed in two references. Please clarify it. The way it is expressed now (as the subject of the sentence) suggests that they are the name of the authors of these references.
  • Line 116. Which reference do the authors refer to with 'Jesus et al.'?
  • Line 128. When the authors say that this method is better that most other methods, what do they refer to? Accuracy? Computing time? Both?
  • Please revise figure 11; there is a textbox which should not appear.
  • Figures 10, 11 and 12 are redundant, as they contain the same information than tables.

Please carefully proofread the paper, as one can find a number of issues with the use of English language.

Reviewer 2 Report

The paper is well organized, easy to follow and the results are clearly presented. Nevertheless, my comments on the paper are the follows:

In the whole chapter 5, the units of the performance have to be explained in the text. e.g. % of what.?

In the figures, the units are missing.

In. Fig. 13 the numbers are very hard to read.

Also, explain what confidence means in your specific case. It makes it easier to read the paper.

Reviewer 3 Report

Interesting approach and nice proposed solution. I found these drawbacks in the paper:

  1. Introduction missing a statement of expectation of this algorithm in comparison with existing ones - it will improve conclusions as the reflection to initial statement.
  2. You provided simulation, nut not real experiment, therefore I would recommend to rethink title.
  3. No formulation for simulation, needs to be explained.
  4. Figure 9 I would recommend to place in amendment and provide little scaled-up, now it prevent reading and brings no real link to your research.
  5. Fig. 10 needs to reformat (columns, there no intermediate values between methods), no legend on y-axis.
  6. There are terms, like "mayavi library" (line 451), which need to be outlines as terms rather then text; there are more places in the text.
  7. Figures 13 and 14 as resulting ones begs for explanation, they are not self-explaining, necessary markers on the figure body.

Round 2

Reviewer 1 Report

The paper has somewhat improved with the revision.

I still think that the paper can be of interest to readers. However, it would be nice if the authors could reconsider including an additional experiment that considers some other classes of the Kitti dataset, in the experimental section, as I stated in my previous report. I understand the point of the authors in their answer to my report. As they say, it is difficult to balance the three goals. Precisely this reason would make it interesting to study which are the limitations of their proposal in this sense.

Additionally:

  • Please check the references in lines 70 and 72 (references 7 and 8).
  • One can also find many errors while referencing equations (e.g. line 227, 239, 308, 313, 328, ...)

Reviewer 3 Report

Here my remarks form the first review really ignore:

  1. fig 9 still in the text and contains low value in this form;
  2. Fig. 10 still has wrong type (interpolation between separate methods on the graph instead of columns, etc.)
  3. There a lot of editing errors - spaces, commas, line spaces, etc.
  4. Results (presented in the fig. 13...15) begs for the explanation, it is not in self explaining form.

Round 3

Reviewer 3 Report

There are a lot of small things left in the formatting, figures captions, etc. Please, do entire edition and read carefully.
